# Parallel Channel Identification and Elimination Method Based on the Spatial Position Relationship of Different Channels

**Mingwei Zhao** [1,2,3,4], **Xiaoxiao Ju** [5,*], **Ni Wang** [1,2,3,4], **Chun Wang** [1,2,3,4], **Weibo Zeng** [1,2,3,4] and **Yan Xu** [1,2,3,4]

[1] Anhui Province Key Laboratory of Physical Geographic Environment, Chuzhou University, Chuzhou 239000, China; zhaomw@lreis.ac.cn (M.Z.); wangni@chzu.edu.cn (N.W.); wangchun@chzu.edu.cn (C.W.); njzwb@chzu.edu.cn (W.Z.); xuyan@chzu.edu.cn (Y.X.)
[2] Anhui Engineering Laboratory of Geo-Information Smart Sensing and Services, Chuzhou 239000, China
[3] Anhui Center for Collaborative Innovation in Geographical Information Integration and Application, Chuzhou 239000, China
[4] School of Geographic Information and Tourism, Chuzhou University, Chuzhou 239000, China
[5] College of Resource, Environment and Tourism, Capital Normal University, Beijing 100048, China
[*] Correspondence: 2230901016@cnu.edu.cn

**Abstract:** Extracting a channel network based on the Digital Elevation Model (DEM) is one of the key research topics in digital terrain analysis. However, when the channel area is wide and flat, it is easy to form parallel channels, which seriously affect the accuracy of channel network extraction. To solve this problem, this study proposes a method to identify and eliminate parallel channels extracted by classical methods. First, the channel level in the study area is marked based on the flow accumulation data, and the parallel channels are then identified using the positional relationship between the different channel levels. Finally, the modification point of the identified parallel channels is determined to eliminate the parallel channels, with the help of the change relationship between the parallel channel and its upper-level channel. In this study, two watersheds in southeast China are selected as examples for method verification and analysis. Experimental results show that the parallel channel identification method proposed in this paper can accurately identify all parallel channels and eliminate the identified parallel channels one by one. The location relationship of the modified channels is consistent with the actual situation, indicating that the proposed method has good application potential in DEM-based channel extraction networks.

**Keywords:** parallel channels; spatial location relationship; DEM; channel level

## 1. Introduction

A channel network is the necessary input data for various hydrological analysis models, which have important applications in related research fields such as soil erosion and rain-flood simulation [1,2]. The channel network is also an important part of the geomorphological feature line, so it plays an important role in the study of landform classification and the analysis of geomorphological evolution process [3,4]. In addition, the channel network is also used as a kind of morphological constraint line in the study of terrain simplification and generalization because it controls the main morphological characteristics of the surface [5–7]. Therefore, it is of great significance to extract the waterway network accurately and quickly.

At present, remote sensing images and DEMs are the main data sources for extracting a channel network. Based on DEM, O'Callaghan and Mark's method of extracting a channel network by surface runoff simulation is widely used because it has a certain theoretical basis and can extract continuous channels [8]. Their method includes the key steps of sink and plane processing, flow direction calculation, flow accumulation calculation and threshold identification. The sink and flat treatment is to ensure that the flow direction of each cell can be calculated. The flow direction identification affects the spatial distribution

of the channel network, and the flow accumulation threshold identifies its initial location. For instance, there are rich research results in the processing of sink and flat [9–14], the determination of flow direction [15–19] and the identification of the threshold parameter of flow accumulation [20–24]. It is worth noting that O'Callaghan and Mark 's method can extract a continuous linear channel network, but processes such as depression filling have actually changed the original topography, which may have adverse effects when conducting hydrological process simulation in the study area [25–27]. Therefore, some scholars have designed the extraction method of channel networks based on hydrodynamics theory in recent years [28]. This method does not require depression filling and has achieved good results in regional hydrological simulations.

Although relevant scholars have carried out valuable research on the key steps of channel network extraction, there are still some problems with the extracted channel network when considering some special terrain, and a typical problem is the parallel channel problem. A parallel channel is actually a parallel (or approximate parallel) channel segment that may occur near the intersection point of the two channels. Parallel channels occur mainly in open and flat areas, and there are two possible reasons for their formation. The first is that the parallel channel is naturally formed in the wide riverbed area due to the existence of local undulating microtopography. The second reason is that for flat riverbed terrain, parallel channels are artificially formed in the flatland processing step during channel extraction. It should be noted that hydrodynamics-based methods can be applied to extract high-resolution fluvial landscapes based on fine-resolution DEM, which is meaningful in hydrological process simulation [29–32]. However, parallel channels need to be eliminated when applied to geomorphological analysis, terrain simplification, etc., because parallel channel has adverse effects on geomorphological analysis and the calculation of related geomorphic parameters on the micro scale.

A comprehensive analysis of current studies on the method of eliminating parallel channels reveals that the research ideas are basically carried out from the perspective of changing the direction of water flow, which is further divided into two ideas: the first is that the DEM is unchanged and the flow direction algorithm is modified [33,34]; the second is that the flow direction algorithm is unchanged and the DEM is modified [35,36]; with the help of other information, the elevation of the local area is fine-tuned to change the incorrect flow direction. Although these studies have achieved ideal results in the extraction of channel networks in the experimental area, their defects are also obvious. For example, whether it is to modify the surface flow algorithm or DEM elevation information, the algorithm is relatively complex. Before these methods are integrated into existing professional analysis software, it is challenging for front-line engineering application personnel to implement them. On the other hand, most of the existing studies on the elimination of parallel channels are aimed at the phenomenon of parallel channels caused by flat land, but the effect of eliminating parallel channels caused by real surface elevation characteristics is not obvious. From the examples of existing studies, it can be seen that in some experimental areas, the method only changes the morphological characteristics of two or more parallel channels, and there are still many parallel channel segments in the large channel. This elimination is essentially ineffective for applications such as channel network density calculation.

To solve the above problems, this study designs a parallel channel elimination method in the DEM-based channel network extraction process for simple operation and obvious effect. This method does not need to change the DEM and the current surface flow algorithm. However, based on the existing methods for extracting the channel network, the parallel channel is identified by the spatial morphology and relative positional relationship between different channel segments. Channel segments are reasonably modified on this basis to eliminate parallel channels. This method is easy to operate, and the linear channel network obtained is more consistent with the channel network on the actual surface. It has certain application value in practical projects such as landform classification, terrain simplification, and DEM generalization.

## 2. Methods and Data

### 2.1. Definition of Parallel Channel

A parallel channel is actually a parallel (or approximate parallel) channel segment that may occur near the intersection point of the two channels. The parallel channel is actually an unreasonable channel section that extended along the main channel because of the flat surface, which prevents the two channels from merging at a proper position. It should be noted that parallel channels do not require strict parallelism between the two channels. What is stressed is that there is an unreasonable extension of the tributary. Figure 1a,b show two typical parallel channels, while Figure 1c shows a normal channel confluence.

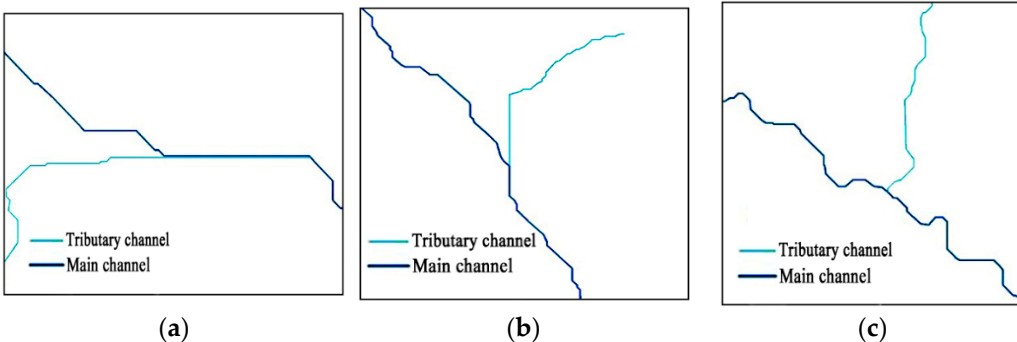

**Figure 1.** Typical examples of parallel channels and normal channels. (**a**) Parallel channel 1. (**b**) Parallel channel 2. (**c**) Normal channel.

In order to identify and eliminate parallel channels, three processes are required. Firstly, the channels in a watershed should be divided into two categories, named the first-level channel and second-channel, respectively; secondly, each second-level channel should be judged if it is a parallel channel by analyzing the spatial relationship of this channel and the first-level channel; and finally, once a second-level channel is marked as a parallel channel, a new channel section should be created in a proper position, and the original section should be deleted. The following Sections 2.2 and 2.3 will introduce the detailed method of the above processes.

### 2.2. Extracting Channel Network and Channel Classification

The main DEM-based channel network extraction method is the surface runoff over-flow model proposed by O'Callaghan and Mark (1984). The basic principle is to identify the flow direction according to the maximum slope between the DEM cell and eight adjacent cells, then calculate the upstream catchment area of each cell and finally identify a catchment area threshold. Cells that are not lower than the threshold are marked as part of the channel. In actual operation, the process of extracting a channel network based on DEM can be divided into four steps: sink and flat surface treatment, flow direction calculation, flow accumulation calculation, flow accumulation threshold setting and extraction channel network. The theory underlying each step and its associated algorithms can be found in the relevant literature; detailed information is not presented in this paper.

Channel networks are spatial data with highly structured features. A channel network usually has various shapes such as network, branch, feather, etc., showing complex geometric features. Depending on the direction of surface flow, channels can be simply divided into mainstream and tributaries. However, when the study area is large and the structure of the channel networks is complex, a classification scheme for the channel network should be made to facilitate the identification of different channel segments. Different levels of channel segments undergo different development processes, and their water quantity, topographic characteristics and vegetation composition are also different. Therefore, research on channel classification methods is also important in the field of hydrology, which plays an important role in channel landform evolution [37], basin soil erosion [38–40], basin vegetation evolution [41,42] and channel biological environment [43].

The existing channel classification models can be classified into two categories: one is based on the relationship between channel nodes and channel reaches, and the other is based on the relationship between mainstream and tributaries. The first representative classification method is Strahler's method [44] (Figure 2a). The basic idea is to define the channel without tributaries as Level 1. The channel level increases after the intersection of two channels. When the two channels at the intersection have the same level, the level increases by 1 after the intersection. Otherwise, if they have different levels, the higher level remains after the intersection. The second representative classification method is Horton's method [45] (Figure 2b). The smallest channel without tributaries is classified as Level 1 in the basic classification process, while only the first level channel is accepted as Level 2, the first and second level channels are accepted as Level 3 and so on.

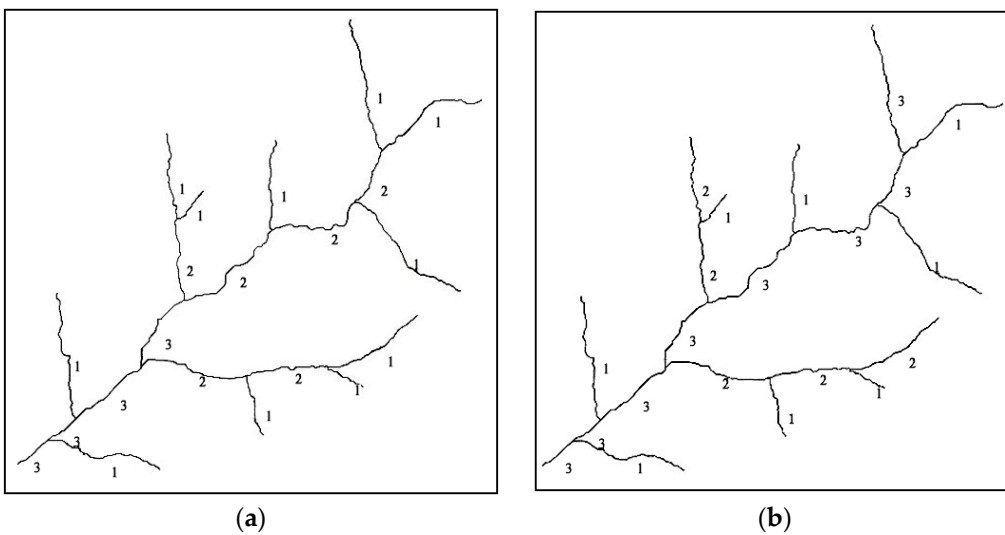

**Figure 2.** Classic channel network classification. (the numbers in the figure represent the channel network level). (**a**) Strahler classification. (**b**) Horton classification.

On the basis of summarizing and analyzing the existing channel network classification methods, this study designs a new channel network level marking method for the convenience of subsequent identification and elimination of parallel channels. The basic idea is to first identify a main channel in the basin and call it the first-level channel; then, all channels that flow into the first-level channel become the second-level channels, all channels that flow into the second-level channel become the third-level channels and so on.

Based on the results of channel network extraction from DEM, the main technical process for channel level identification is as follows:

1.  Identify the first-level channel. The cell where the outlet of the basin is located is taken as the first cell of the first-level channel (Figure 3a), and the second cell of the first-level channel is then found by searching the area around it for the cell with the highest flow accumulation value. As shown in Figure 3b, the flow accumulation value of the cell labeled g1 is greater than that of the cell labeled g2, so cell g1 is used as part of the current channel. By analogy, a new cell is searched around the newly identified first-level channel cell until the flow accumulation of surrounding adjacent cells is lower than the threshold set in the process channel extraction. Then, the first-level channel has been identified (Figure 3c).

2.  Searching for the cell with the greatest flow accumulation in addition to the cells that have completed the level marking, which must be contiguous to the marked cells, to be marked as the next level of channel segment (Figure 3d), assuming that during the search, the cells labelled g1, g2 and g3 all have the largest flow accumulation close to the marked cells, and we need to select the largest cell (g1) among g1, g2 and g3 to be marked. It should be noted that a cell cannot be marked more than once throughout

the process. Assuming that its level is N, then the cell is the first cell in a new channel, and its level is N + 1. The channel is identified by searching for cell units that meet the conditions around it with reference to the method in (1) (Figure 3e).

3.  Step (2) is repeated continuously until there is no cell larger than the threshold of flow accumulation in the unmarked cells. The channel level identification is thus completed, and all channels in the test area have been marked (Figure 3f).

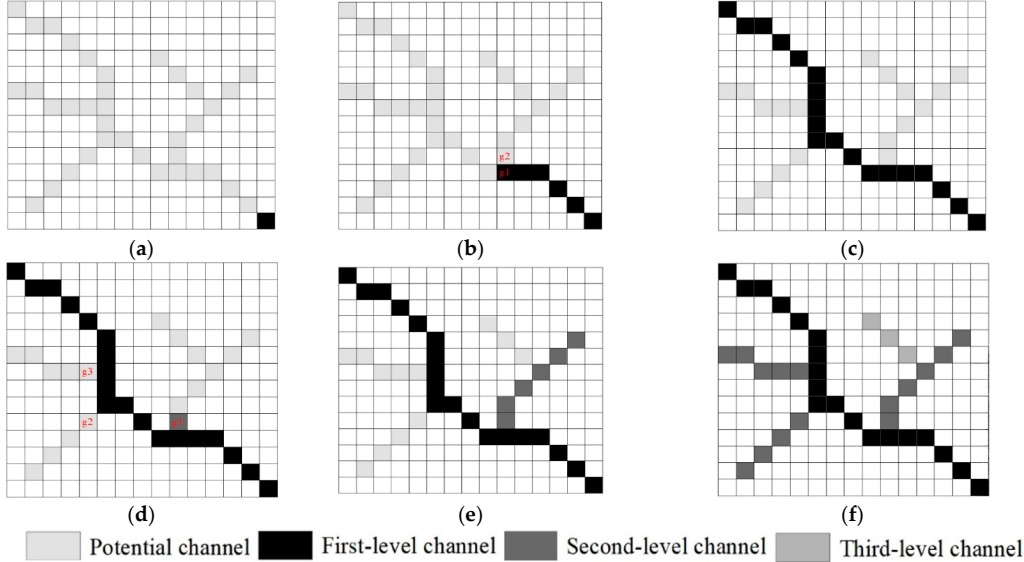

**Figure 3.** Channel classification process. (**a**) Step 1. (**b**) Step 2. (**c**) Step 3. (**d**) Step 4. (**e**) Step 5. (**f**) Step 6.

Figure 4 shows the new channel network classification results with the same example as Figure 2. It can be seen that the compared with the classification scheme shown in Figure 2, the main difference of the new classification scheme is that there is only one first-level channel, which is very convenient for judging the second-level channels one by one.

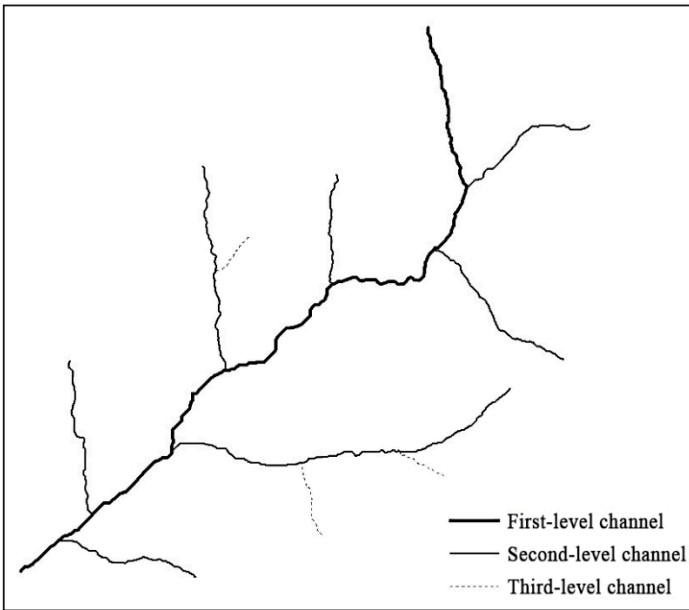

**Figure 4.** Example of channel level identification in this study.

### 2.3. Identification and Elimination of Parallel Channels

This section uses the three channel types shown in Figure 1 as examples to introduce methods for identifying and eliminating parallel channels. From the spatial location relationship between first-level and second-level channels, the second-level channel can be regarded as the process in which water flows from its starting position to the first-level channel and finally flows into the first-level channel. In this process, under normal circumstances, the distance between the water flow and the first-level channel should be closer and closer until it enters the first-level channel, and the distance becomes 0. However, when there is a parallel channel, the distance between the water flow and the first-level channel decreases first; then, there is a constant (or approximately constant) process, and the distance finally gradually decreases to 0. Therefore, the shortest distance from a point on the second-level channel to the first-level channel can be used as an index to describe the spatial relationship of the two channels, and this index (S1) can be computed as follows:

$$S1 = \min \lceil P_i P_j \rceil \tag{1}$$

in which $P_i$ is the point in the second-level channel, $P_j$ is the point in the first-level channel and $\lceil P_i P_j \rceil$ means the distance of the two points.

At the same time, when calculating S1 from the intersection point of the two channels, the distance between the corresponding points of S1 in the first-level channel and the intersection point is continuously changing (Figure 5). For the parallel channels, in the calculation of S1, when point P uniformly moves away from the intersection point O, the projected point P' of point P is far away from the intersection point O at a relatively fast speed in the unreasonable channel section, and the change speed of point P' from the intersection point will slow down when the unreasonable channel section is past. Therefore, the distance from point P' to the intersection point O can be used as the second parameter to describe the spatial relationship of the two channels, which can be named as S2.

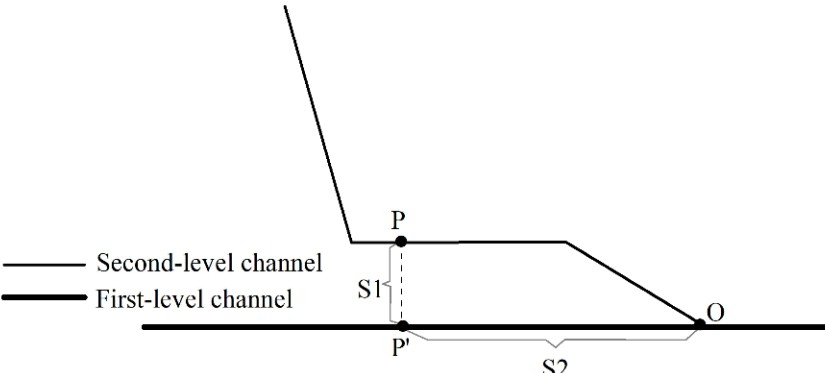

**Figure 5.** The diagram for calculating S1 and S2.

Taking the three types of channels shown in Figure 1 as examples, it is possible to analyze whether S1 and S2 can successfully identify parallel channels. From the intersection point of the two channels, S1 and S2 at different positions are calculated with a unit as the step size, and the curves of S1 and S2 are drawn (Figure 6a,b). For the curves of S1 (Figure 6a), it can be seen that if there is a channel section in the second-level channel that closely parallels the first-level channel (Figure 1a), S1 maintains a constant during the parallel section and increases rapidly when leaving the parallel section, so S1 can be used to identify the parallel channel in this case. However, S1 also increases in the unreasonable channel segment if the second-level channel is not strictly parallel to the first level channel (Figure 1b), and the trend of S1 variation is similar to the normal channel in this case. In other words, S1 cannot be applied to identify the parallel channels similar to Figure 1b.

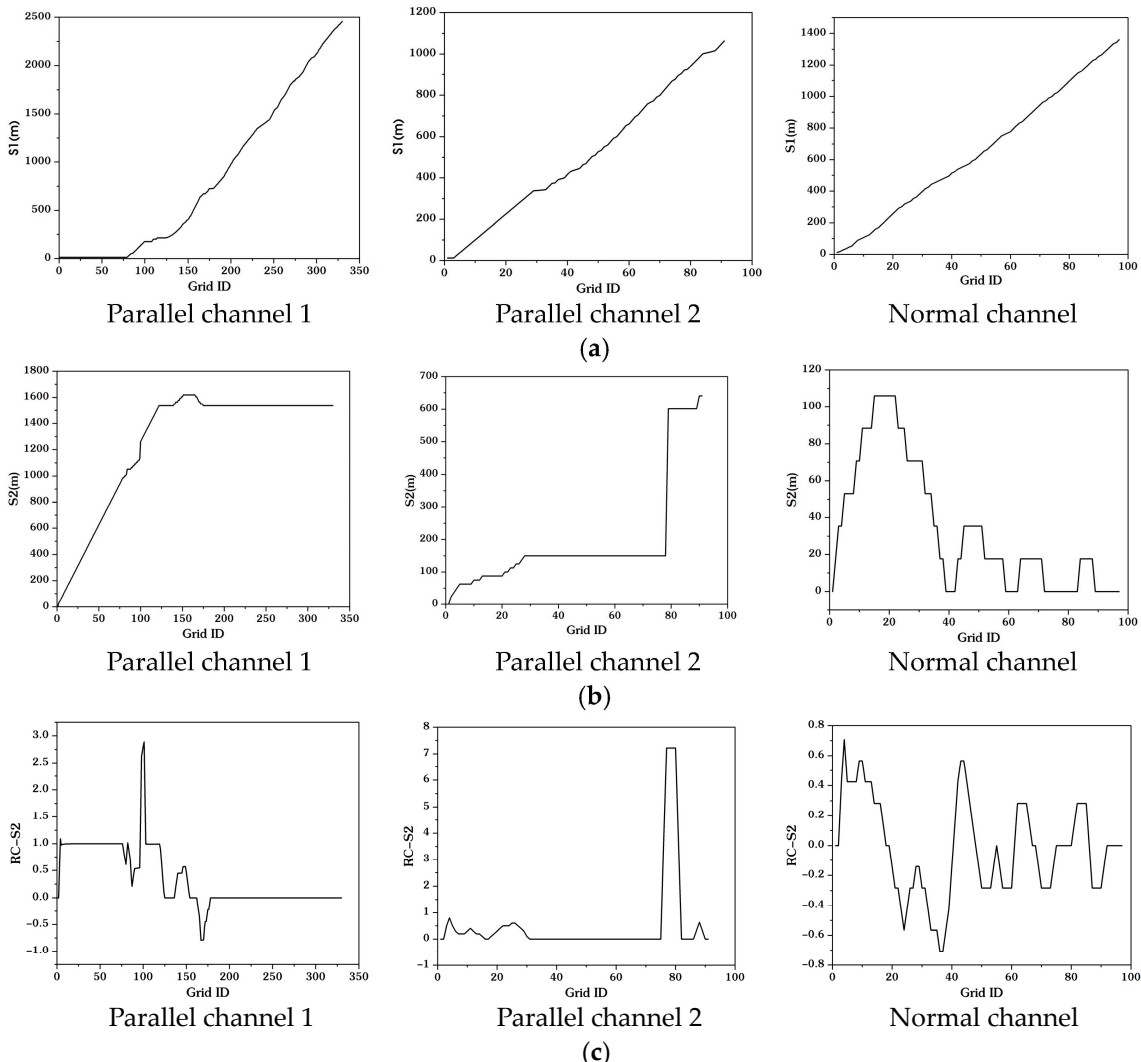

**Figure 6.** The curves of S1, S2 and rate of change of S2 for different channels. (**a**) The curves of S1. (**b**) The curves of S2. (**c**) The rate of change of S2.

For curves S2 (Figure 6b), it can be seen that if a second-level channel has an unreasonable segment (Figure 1a,b), S2 first increases from the point of intersection of the two channels and finally tends to be a constant, which is approximately equal to the distance of parallel river extension. As for the normal channel (Figure 1c), S2 increases over a short distance and then oscillates rapidly around 0. In addition, the rate of change of S2 can be calculated (Figure 6c). It can be found that compared with normal channels, the change rate of S2 near the intersection point of parallel channels is much larger, so such a characteristic can be used to distinguish parallel channels from normal channels.

It is important to note that parallel channels are unreasonable channel segments that occur near the intersection point of two channels. However, for a complex channel network, a section of a second-level channel may form an approximately parallel channel with a section of the first-level channel away from the intersection point (Figure 7), which is obviously not the parallel channel we are studying. Therefore, in order to avoid interference of channels similar to Figure 7 on the final experimental results, the second-level channel should be limited to a certain range when calculating S2 and its rate of change. Observing the parallel channels in Figure 1 once again, it can be found that the extending direction of the second-level channel changes significantly at the end of the parallel section, so this position can be taken as the last point (abbreviated as DC-Point in this paper) to calculate S2 and its rate of change. It is easy to calculate the position of this point: first, connect the

beginning and end points of the analyzed second-level channel to generate a line. The required point can be determined by calculating the maximum vertical distance from the intermediate point to the line (Figure 8).

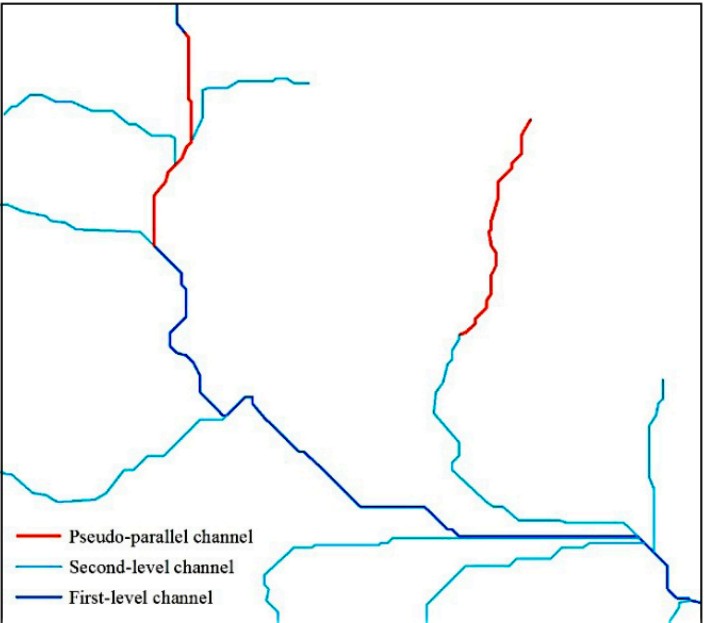

**Figure 7.** Pseudo-parallel channel.

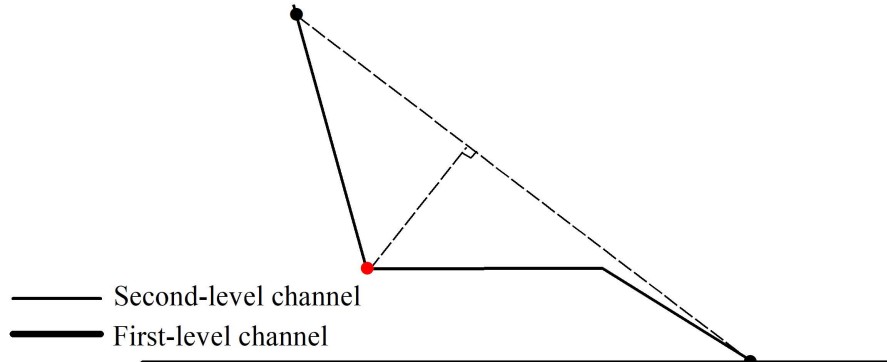

**Figure 8.** The diagram of the DC-Point (the red solid circle denote the DC-Point, the dashed line is is the extension to the second-level channel from the intersection of the first-level and second-level channel used to find the DC-Point).

The DC-Points of the three channels shown in Figure 1 were computed, and the curves of the rate change of S2 were redrawn (Figure 9). And then a parameter named RCD can be defined to identify parallel channels as follows:

$$\text{RCD} = \sum_{i=1}^{n} rc_i \tag{2}$$

in which $rc_i$ is the rate of change of S2 at each point, and $n$ is the total number of points between the intersection point and the DC-Point. The RCDs of the three channels in Figure 1 are 0.97, 0.78 and 0.16, respectively. It can be found that the RCD of the parallel channel is obviously larger than the value of the normal channels, indicating that this parameter can be used to identify parallel channels.

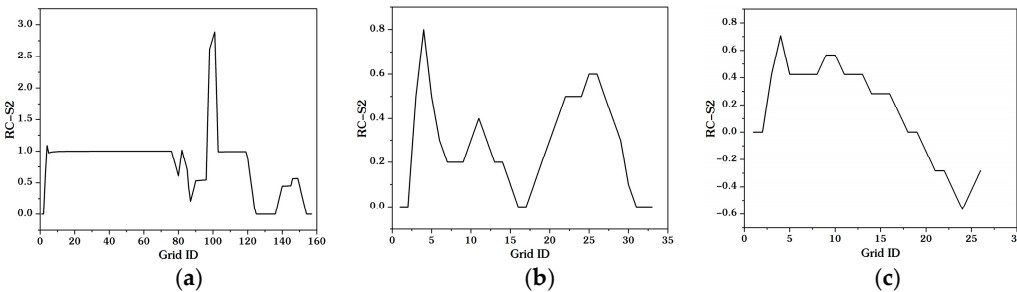

**Figure 9.** The rate of change of S2. (**a**) Parallel channel 1. (**b**) Parallel channel 2. (**c**) Normal channel.

The next step is to correct unreasonable channel segments after identifying parallel channels. It is easy to deduce that the DC-Point in the second-level channel is also the position that needs correction. Then, the distances from the DC-Point to different vertices on the first-level channel are calculated, and the point in the first-level channel corresponding to the shortest distance is the new intersection point of the two channels. The corrected channel section can be created by connecting the DC-Point and the new intersection point (Figure 10).

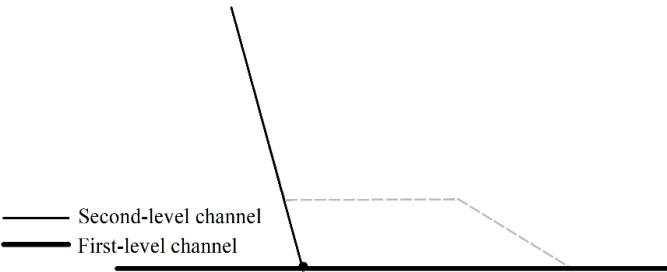

**Figure 10.** The corrected channel section.(the dashed line parallel channel segment).

In general, the elimination of the parallel channel is completed by deleting the channel segment between its modification point and the intersection point with the upper channel. However, prior to the above process, it is necessary to verify whether there are other channels that intersect with the channel segments to be deleted. In this case (as shown in Figure 11a), it is necessary to take the vertical line from the intersection point between the two to the upper channel as the modified channel (as shown in Figure 11b) and then delete the channel segments marked to be deleted.

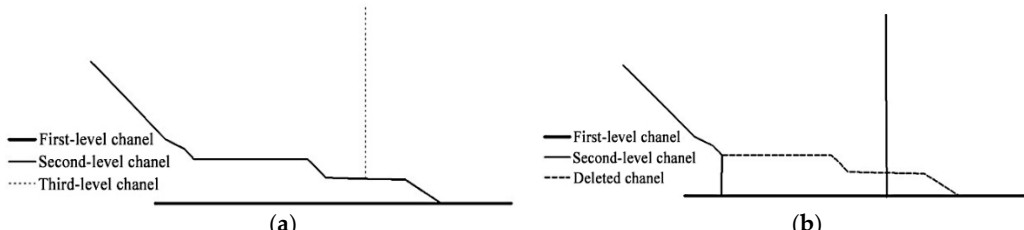

**Figure 11.** Deleting process of parallel channel. (**a**) Parallel channel before deletion. (**b**) Parallel channel after deletion.

According to the above method, all channels marked as parallel channels are processed one by one, and the elimination of parallel channels in the analysis area is completed. The basic workflow for eliminating parallel channels based on DEM is shown in Figure 12, which includes the main key processes, such as channel level determination, parallel channel elimination and so on. Throughout the process, the initial channels are produced with ArcGIS 10.2 software, while the other processes are performed with VS2010-based programs written by the authors.

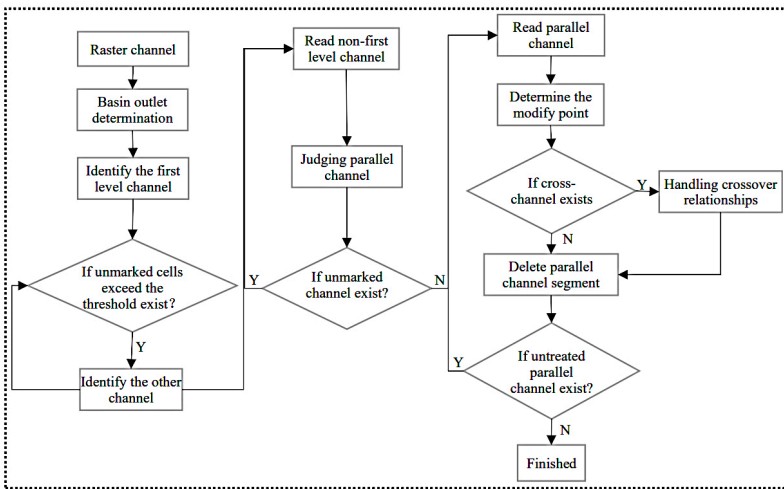

**Figure 12.** The flowchart for parallel channels identification and elimination.

## 2.4. Test Area and Data

The test area selected in this paper is located near the boundary of Longquan City and Suichang County in Lishui City, Zhejiang Province. The whole area is located in the middle and upper reaches of the Oujiang River in Zhejiang Province, which is typical mountain terrain, and the surface elevation of the area varies from 183 m to 1516 m (Figure 13). Two small watersheds of the Jinshuitan Reservoir in the test area were selected for the channel network extraction and analysis. The main channels in the two watersheds belong to a U-shaped valley in surface morphology, while the tributary channels belong to a V-shaped valley, and the parallel channels are mainly distributed in the lower reaches of the watersheds where the riverbed landform is well developed.

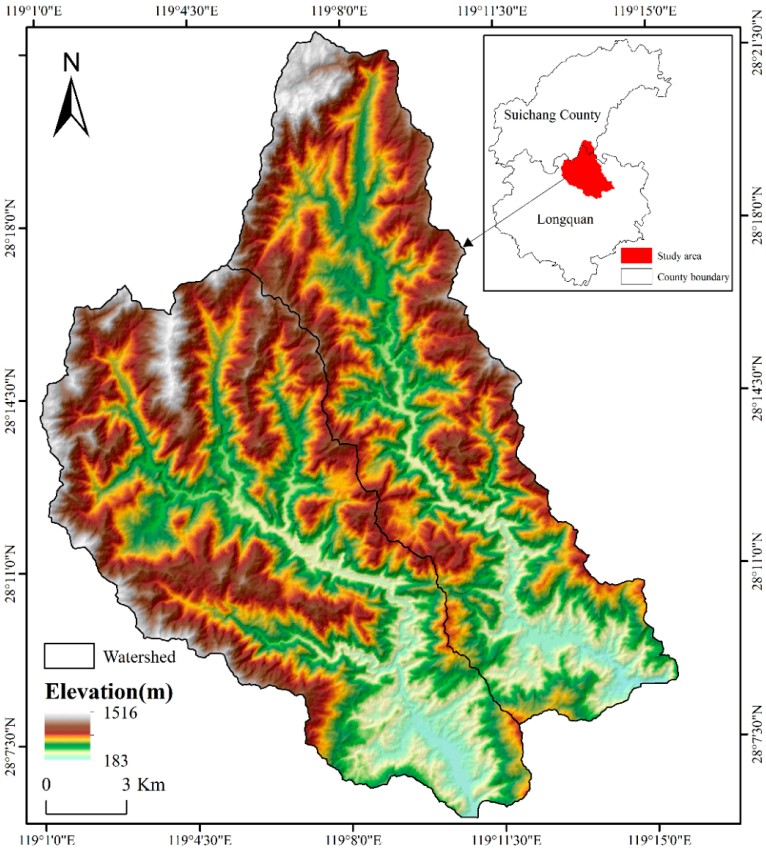

**Figure 13.** Study area.

The DEM data used for the experiments in this paper are the data products of ALOS PALSAR released in 2011, which were obtained by the ASF's radiometric terrain correction project to correct geometry and radiometry of the Phased Array L-band Synthetic Aperture Radar (PALSAR) carried out by the ALOS Advanced Earth Observation Satellite, jointly developed by METI and JAXA in Japan and distributed to the public free of charge. The data download address is the EarthData Search platform (the data accessed on 22 July 2010, https://earthdata.nasa.gov/), with a cell size of 12.5 m, using the WGS84 coordinate system and EGM96 elevation data.

## 3. Results Analysis

### 3.1. Extraction and Classification of the Channel Network

In this study, the channel network of test watersheds was extracted from DEMs using the method proposed by Callaghan and Mark (1984). It is well known that the determination of the flow accumulation threshold is a critical step in the channel network extraction process, and a heuristic idea is applied in this study to determine the most appropriate value. The main objective of this idea is to extract the first-level and second-level channels because parallel channel sections primarily exist near the intersection point of the first-level and second-level channels.

Based on the extracted channel network, the channels of the two test small watersheds are graded and marked using the classification method described in Section 2.2. The results are shown in Figure 14, it can be found that for test watersheds, the first-level channel extends from the outlet to the top of the small watershed, and the second-level channels are distributed on both sides of the first-level channel. The spatial relationship between the two level of channels is clear, which can well support the subsequent parallel channel identification and elimination process.

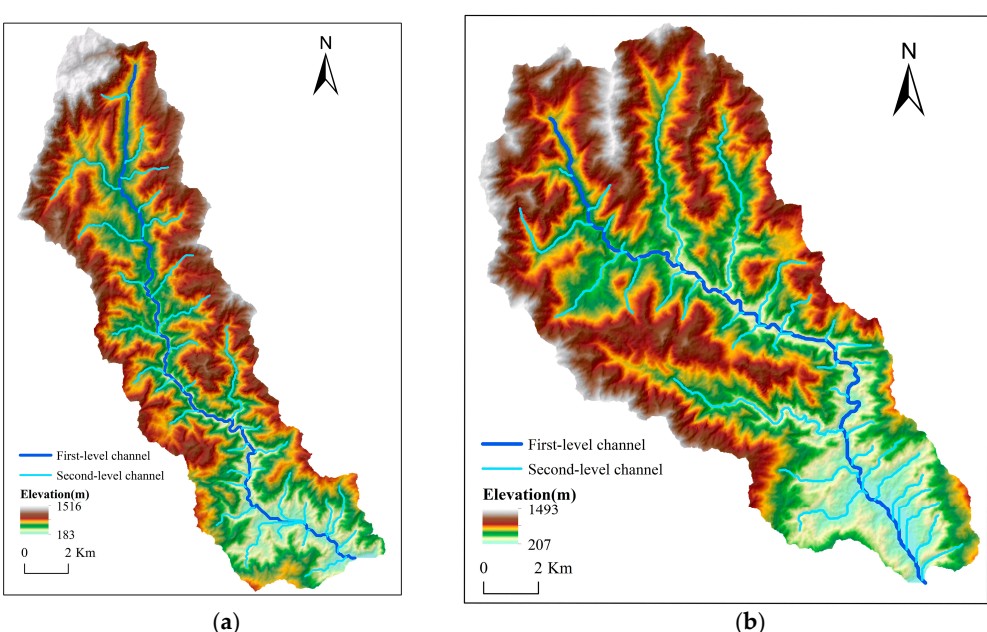

**Figure 14.** The classification of channel network. (**a**) Watershed 1. (**b**) Watershed 2.

### 3.2. Identification and Marked of Parallel Channels

For the selected test watersheds, the RCD of all the second-level channels was calculated one by one. Figure 15 shows the RCD scatter plots in the two watersheds, in which the horizontal scale is the serial number of the second-level channels, and the vertical scale is the RCD. The next work is to identify parallel channels from scatter plots. In other words, a threshold of the RCD value is required to divide the channels into two groups, one being parallel channels and the other being normal channels. There are two test watersheds in this

study, so our strategy is to determine the threshold of the first watershed by the statistical method, and the second test watershed was taken as the validation area of the threshold.

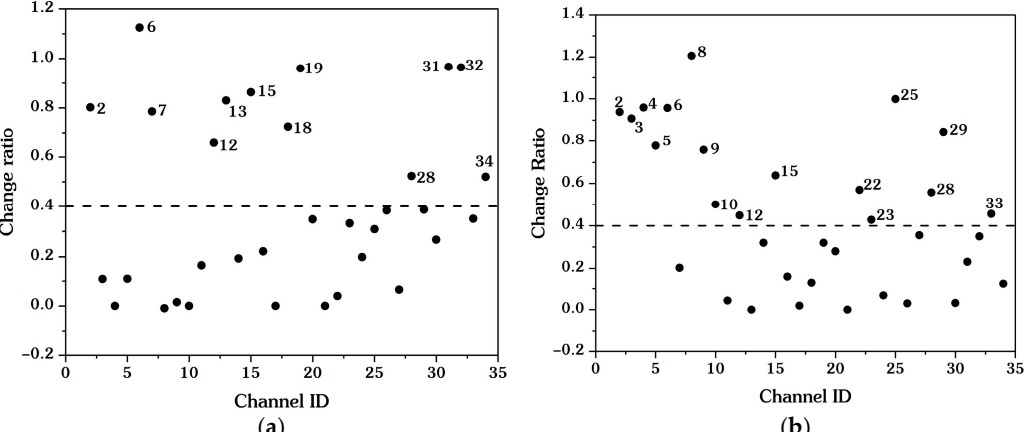

**Figure 15.** The judgement of parallel channels. (**a**) Watershed 1. (**b**) Watershed 2. (The dashed line serves as a threshold to distinguish between normal channels and parallel channels. Channels with a change ratio than the threshold are considered parallel channels, while those with a rate lower than the threshold are considered normal channels).

An initial threshold is set to 0.8 for the first watershed, and then it was checked whether the second-level channel whose RCD is greater than the threshold is a parallel channel one by one. If this is the case, the initial threshold is reduced by 0.1, and channels with an RCD greater than the new threshold are examined. The above process is continued to run until there is at least one normal channel's RCD is larger than the current threshold, and then the former threshold is taken as the final threshold. In this study, the final threshold for the first watershed is 0.4, and the threshold for the second watershed is also set to 0.4. Figure 16 shows the parallel channels of the two watersheds at the same threshold. There are 12 parallel channels in the first watershed and 16 parallel channels in the second watershed, the ratio of the parallel channels in the channel network is 36% and 48% in the two watersheds, respectively, indicating that the parallel channel is ubiquitous in this region.

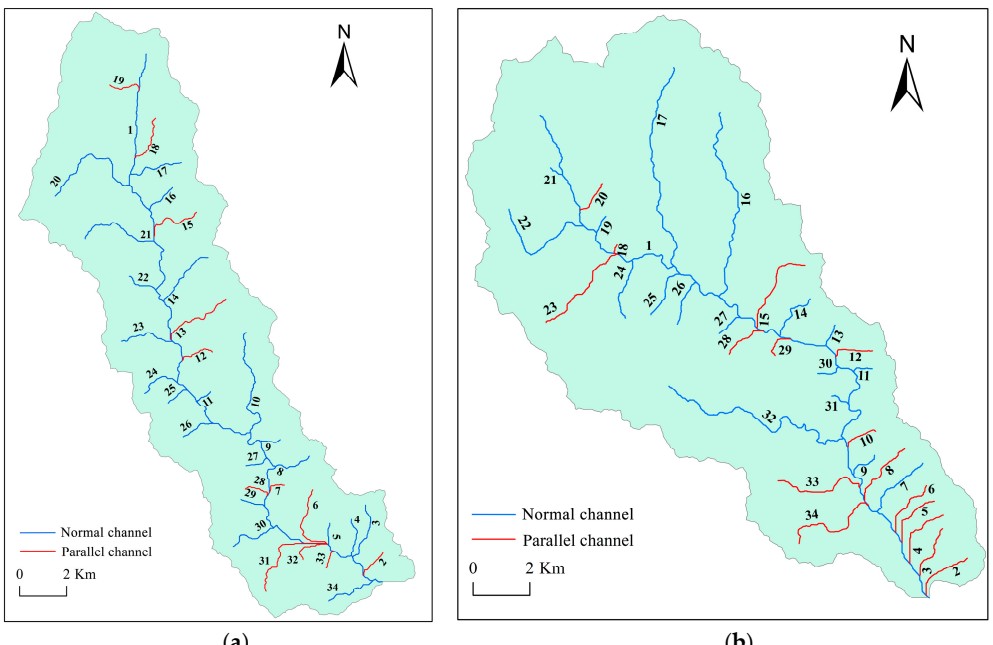

**Figure 16.** Markings of parallel channel. (**a**) Watershed 1. (**b**) Watershed 2. (the numbers are channels's ID).

In order to test the accuracy of this method, all the second-level channels of the two watersheds were inspected by hand vision. The inspection results show that all marked channels are parallel channels. That is, the method does not misjudge normal channels as parallel channels. However, there are some parallel channels that could not be identified, one of which is shown in Figure 17a. There are two reasons for the omission of such channels. The first is that the parallel section is too short, and the other is that the first-level channel and the second-level channel are not strictly parallel. By way of comparison, Figure 17b shows an example where although the unseasonable section is very short, it is still successfully identified because the unseasonable section is completely parallel to the first-level channel. There are two channels that could not be identified in the two watersheds, respectively, similar to the channel in Figure 17a. Therefore, considering that this omission has little impact on the entire channel network, the parallel channel identification method designed in this research is accurate and effective.

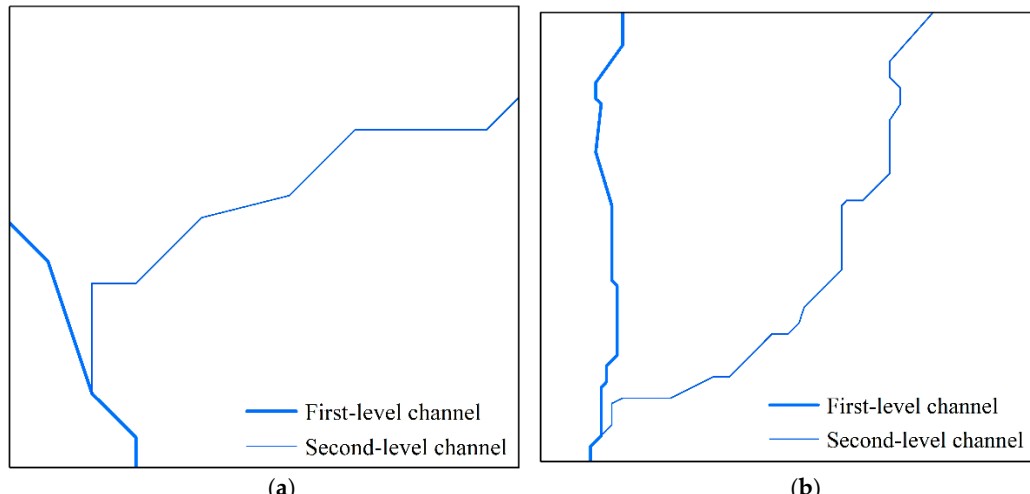

**Figure 17.** Identification status of parallel channel in watershed 1. (**a**) Unidentified (ID 3). (**b**) Identified (ID 18).

### 3.3. Elimination of Parallel Channels

After identifying the parallel channels in the studied watershed, the next step is to eliminate the unreasonable segments of the marked parallel channels. First, the positions of the modification points of the second-level channels marked as parallel channels are calculated. Figure 18 shows the spatial location of modification points of typical parallel channels in two test watersheds, where 'DC-Point' represents modification points on the second-level channels and 'New intersection point' is the corresponding modification points on the first-level channel. Once the location of the modification points is determined, a new channel section can be created by connecting the modification points of the two level channels. The original channel section between the modification point of the second-level channel and the intersection point of the two level channels should be deleted at the same time, and it can be found that the modification point of the first-level channel becomes the new intersection point.

The modified channel network of the two test watersheds is shown in Figure 19, where the small figure is the correction result of the area with obvious parallel channels. It can be seen from the figure that the modified channels are very reasonable in terms of spatial form and extension direction, and all the identified parallel channels can be reasonably modified, indicating that the identification and elimination method designed for dealing with parallel channels in this study is reasonable and effective.

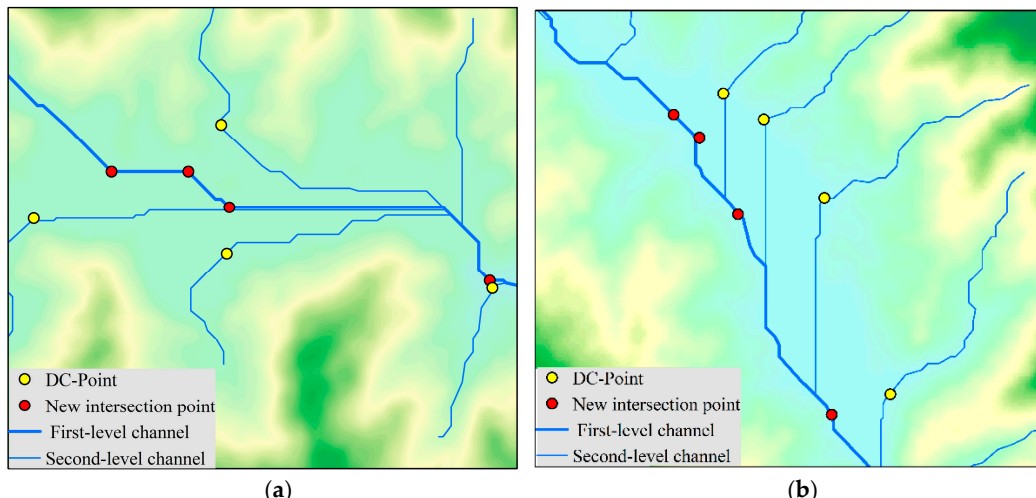

**Figure 18.** The spatial location of the modification points. (**a**) Watershed 1. (**b**) Watershed 2.

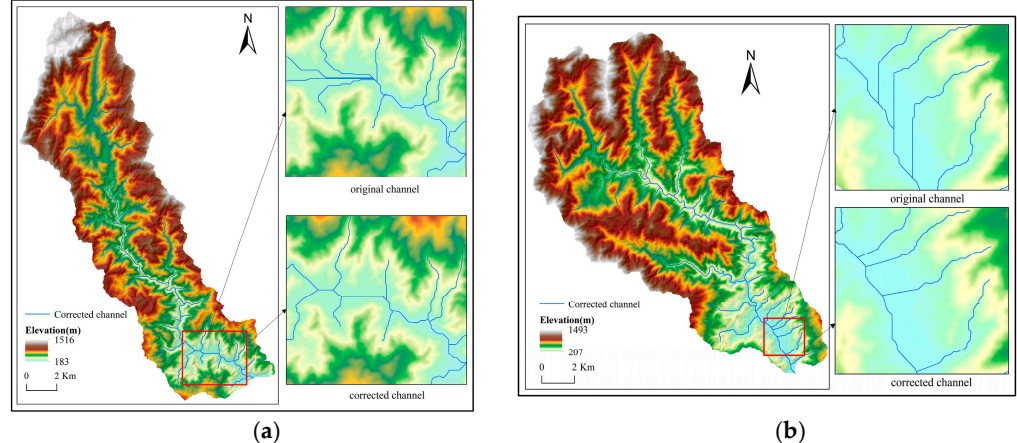

**Figure 19.** The modified channel network of the two test watersheds. (**a**) Watershed 1. (**b**) Watershed 2.

## 4. Discussions

### 4.1. Effluence on Channel Length

In areas with severe water erosion, the total channel length or channel density of the area is an important indicator to measure the development process of eroded landform. Therefore, it is very important to calculate the channel length accurately to identify the development process of the landform. This section compares the length variation of each channel before and after the elimination of parallel channels.

The channel network becomes more rational in spatial relationships and patterns after eliminating parallel channel segments, and this process also leads to variation of channel length of marked parallel channels. Table 1 shows the changes in the length of marked parallel channels in the two test watersheds, where Difference Value (DV) represents the difference between the original length and the corrected length, and Ratio Value (RV) represents the ratio of the above difference to the corrected channel length. It can be seen that the changes of the parallel channel length show some common characteristics in the two watersheds. The first feature is that the variation of RV is very large, and the maximum value reaches 105% in the first watershed, indicating that the length of the deleted channel is greater than the length of the corrected channel, and the minimum is only 2%; in the second watershed, the maximum is 36%, and the minimum is only 1%. The second feature is that the obvious parallel channels with larger DV values in the table are distributed in the downstream region of the first-level channel, mainly because the downstream valley is wide and flat, which makes it easier to form parallel channels.

**Table 1.** The changes in the length of marked parallel channels.

| Watershed 1 | | | Watershed 2 | | |
|---|---|---|---|---|---|
| ID | DV | RV | ID | DV | RV |
| 2 | 191 | 16% | 2 | 293 | 15% |
| 6 | 893 | 36% | 3 | 128 | 6% |
| 7 | 226 | 33% | 4 | 631 | 34% |
| 12 | 119 | 8% | 5 | 409 | 22% |
| 13 | 226 | 7% | 6 | 269 | 13% |
| 15 | 443 | 19% | 8 | 196 | 8% |
| 18 | 68 | 3% | 10 | 89 | 7% |
| 19 | 179 | 13% | 12 | 93 | 6% |
| 28 | 70 | 6% | 15 | 260 | 8% |
| 31 | 1480 | 50% | 18 | 39 | 11% |
| 32 | 884 | 105% | 20 | 52 | 4% |
| 33 | 37 | 5% | 23 | 155 | 4% |
| | | | 28 | 194 | 13% |
| | | | 29 | 310 | 36% |
| | | | 33 | 32 | 1% |
| | | | 34 | 142 | 4% |

From the perspective of the total length of the channel network, the total channel length in the first watershed decreased by 4914 m after the elimination of parallel channels, and the total channel length corrected is 99,148 m. If the corrected channel network is taken as the actual one, parallel channels can be found to increase the total length by 4.96%. It is important to note that 4.96% does not necessarily indicate the improvement percentage, as not all channel networks in a watershed consist of parallel channels. Typically, parallel channels are found near the junction of the first level and the second-level channels in relatively flat valleys (see Figure 1a). Therefore, the observed improvement is quite significant for the entire channel network in the watershed. For the second watershed, the total channel length decreased by 3309 m after the elimination of parallel channels, the total channel length corrected was 107,679 m and the parallel channels resulted in a total length increase of 3.10%. It can be seen that it is very necessary to identify and eliminate parallel channels for calculating the total length or density of channels in a certain area.

### 4.2. Effluence on the Structure of Channel Network

Parallel channels not only affect channel length, but also vary the topology of the channel network. Figure 20 shows the Strahler classification of the channel network near the outlet of the first watershed. It can be seen that the elimination of parallel channels not only changes the spatial form of the channel network, but also changes the topological structure of the channel network. Taking Figure 20 as an example, the original extracted channels can be divided into three levels, and the channels after eliminating parallel waterways are divided into two levels. This is because in the original waterway network, the channels that should be merged into the main channel are merged into other channels, and the method of dealing with this situation is given in Section 2.3 (Figure 11).

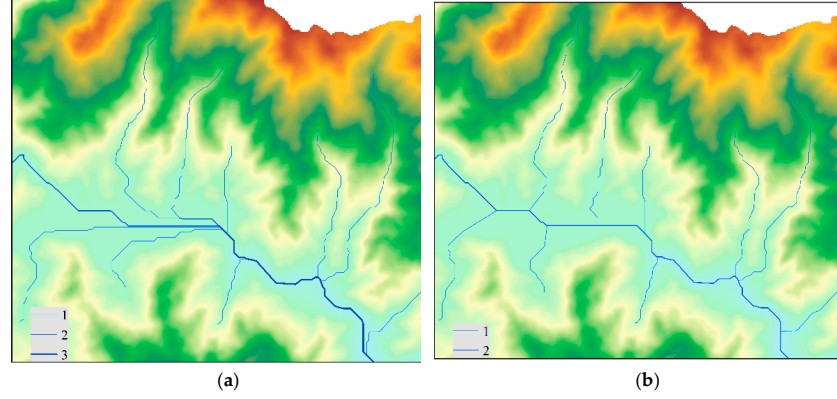

(**a**)         (**b**)

**Figure 20.** Classification of the original and corrected channel using Strahler's method. (**a**) Original channel. (**b**) Corrected channel.

## 5. Conclusions

When extracting a DEM-based channel network, parallel channels are easily formed for wide and flat channels, which affects the accuracy of channel network extraction and brings problems for subsequent application analysis. To solve this problem, this study designed a parallel channel identification and elimination method using the spatial location relationship between different channels and selected a typical experimental area for experimental analysis. The main conclusions of this paper are as follows:

1. The channel classification method defined in this study has a clear concept and a simple algorithm. It can mark different channel levels based on flow accumulation data and can be useful for subsequent operation of parallel channel identification and elimination.

2. The parallel channel identification method designed in this study is based on the changing characteristics of the spatial location relationship between different channel levels. It was able to accurately identify almost all the parallel channels in the study area, and some channels may appear to be parallel channels that were not identified because the parallel section to the upper channel was too short.

Finally, the parallel channel elimination method proposed in this study is actually carried out on the current classical channel extraction results. It does not involve modifying the original data and optimizing key algorithms in the channel extraction process. The whole process is simple and easy to implement, and it has good application potential for scientific research and engineering applications in related fields such as soil and water conservation, geomorphic parameter calculation and so on.

Of course, there are still some shortcomings in this study. The proposed method only addresses the channel network and does not address other surface confluence processes, such as watershed division. In the process of surface confluence, the watershed division of the study area is carried out based on the extracted channel network. Therefore, the corresponding watersheds should also be modified after the elimination of parallel channels. However, this study does not address this issue because this work includes modifications of the DEM itself. The author of this paper has done some exploratory work to optimize the flow path on the surface of urban roads [46]. The modified urban roads' DEM can ensure that the flow path conforms to the actual situation, and the watershed division is indisputable reasonable in this case. However, the shape and function of the urban road are special, so the DEM modification method was designed for this type of surface. Therefore, the related methods cannot be directly used in the study of natural terrain, and how to deal with natural terrain to obtain reasonable watersheds is the author's follow-up work.

**Author Contributions:** Conceptualization, Mingwei Zhao and Xiaoxiao Ju; formal analysis and data curation, Mingwei Zhao; methodology and supervision, Xiaoxiao Ju; writing—original draft, Mingwei Zhao and Ni Wang; visualization and investigation, Weibo Zeng; writing—reviewing and editing, Xiaoxiao Ju, Chun Wang, Yan Xu and Weibo Zeng. All authors have read and agreed to the published version of the manuscript.

**Funding:** This research was supported by Anhui Province Key Laboratory of Physical Geographic Environment, P.R. China (2023PGE003); Anhui Province Universities Outstanding Talented Person Support Project (No. gxyq2022097, No. gxyq2021217); Major Project of Natural Science Research of Anhui Provincial Department of Education (No. 2022AH040150, No. KJ2021ZD0130); Key Project of Natural Science Research of Anhui Provincial Department of Education (No. KJ2021A1075, No. KJ2021A1080); The guiding plan project of Chuzhou science and Technology Bureau (No. 2021ZD008); "113" Industry Innovation Team of Chuzhou city in Anhui province.

**Institutional Review Board Statement:** Not applicable.

**Informed Consent Statement:** Informed consent was obtained from all subjects involved in the study.

**Data Availability Statement:** The raw DEM data that were processed in this study are available from Geospatial Data Cloud (the data accessed on 22 July 2010, http://www.gscloud.cn). All datasets used and analyzed during the current study are available from the corresponding author (2230901016@cnu.edu.cn) on request.

**Acknowledgments:** We are thankful for all of the helpful comments provided by the reviewers.

**Conflicts of Interest:** The authors declare no conflicts of interest.

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
