# Peer review of "Parallel Channel Identification and Elimination Method Based on the Spatial Position Relationship of Different Channels"

_ijgi, doi:10.3390/ijgi13010013_

Round 1

Reviewer 1 Report

Comments and Suggestions for Authors

I have previously reviewed this manuscript for another journal. In my previous review, I have requested the authors to make their algorithm code publicly available (e.g. Github repository) so that other people (including reviewers) can validate the algorithm and possibly use the code in the future. This will benefit the entire community. The open source tradition has been followed by many researchers in this research field and the authors should follow this tradition as well. The authors, however, refused this request. Therefore, I suggest to reject the previous submission. In this review, I am still requesting the authors to release their code before I make further assessment on the current manuscript.

Comments on the Quality of English Language

The English writing is fine.

Author Response

Thank you for your comments, we have provided the executable programme and test data of the manuscript according to your suggestions. For ease of use, we have encapsulated certain core codes. Additionally, we have provided an operation guide for the program. Please find the program file in the submission system for your reference. 

Reviewer 2 Report

Comments and Suggestions for Authors

This paper could be accepetd after the authors address the minor comments in the attached annotated manuscript. 

Comments on the Quality of English Language

This is a well organized manuscript. 

Author Response

Thank you for your feedback. We have responded to your comments in the annotated PDF file as requested. Please check the attachment for our response.

Reviewer 3 Report

Comments and Suggestions for Authors

The manuscript presents an algorithm to identify and remove parallel river segments from channel networks derived from DEMs. Parallel rive segments widely exist in drainage networks if no special treatments are applied. The study helps to enrich the literature on drainage network extraction from DEMs, and it is an interesting experiment for digital terrain analysis. Here are some comments to the authors.

1. Parallel segments can appear on any location. Why does the algorithm only consider the second-level and the first level channel?

2. The quality of the figures in the manuscript is inadequate, especially Figures 6 and 9.

3. Figure captions must be descriptive and illustrative. The caption of figure 20(line 444, p16) should be modified from " The Strahler topology of channel network in the first watershed " to " Classification of the original and corrected channel using Strahler’s method ".

Comments on the Quality of English Language

The English writing for the entire manuscript needs to be carefully checked and improved. For instance, there are some grammatical errors in the sentencethis idea’s main object is to extract the first-level and second-level channels because the parallel channel sections have mainly existed near the intersection point of the first-level and second-level channels.”(line 327-329, p11), and it should be rephrased asthe main objective of this idea is to extract the first-level and second-level channels because parallel channel sections primarily exist near the intersection point of the first-level and second-level channels.

Author Response

Thank you for your feedback. Below are responses to each of your comments:

1. Parallel segments can appear on any location. Why does the algorithm only consider the second-level and the first level channel?

Response: Thank you very much for your suggestion. The so-called first-level channel and second-level channel are different marks made by the authors for the channels in the study area. Through a large number of experiments, we found that the parallel channels mainly occurred in a rather wide valley, which corresponds to the intersection of the first-level channel and the second-level channel in the study area. If the study area is large enough and the channels are divided into more levels, then the parallel channel may also appear at other channel levels. In this case, the treatment method is the same.

2. The quality of the figures in the manuscript is inadequate, especially Figures 6 and 9.

Response: Thank you very much for your suggestion. Regarding the issue of image quality that you mentioned, we found that images are compressed when they are inserted into Word. Therefore, when submitting the manuscript to the system, we have separately submitted each image one by one.

3. Figure captions must be descriptive and illustrative. The caption of figure 20(line 444, p16) should be modified from " The Strahler topology of channel network in the first watershed " to " Classification of the original and corrected channel using Strahler’s method ".

Response: Thank you for your suggestions. The manuscript has been revised in accordance with your recommendations.

Comments on the Quality of English Language

4. The English writing for the entire manuscript needs to be carefully checked and improved. For instance, there are some grammatical errors in the sentence“this idea’s main object is to extract the first-level and second-level channels because the parallel channel sections have mainly existed near the intersection point of the first-level and second-level channels.”(line 327-329, p11), and it should be rephrased as“the main objective of this idea is to extract the first-level and second-level channels because parallel channel sections primarily exist near the intersection point of the first-level and second-level channels.”

Response: Thank you for your suggestions. We have carefully reviewed the manuscript and made revisions based on your advice.

Reviewer 4 Report

Comments and Suggestions for Authors

This article “Parallel channel identification and elimination method based on the spatial position relationship of different channels”aims to Identify and eliminate the parallel channels of river network extracted from DEM. It is a interesting experiment for digital terrain analysis.

So I think this work is of great value in studies about ditital teraain analysis, and of course it must be helpful to enhance the influence of this journal. So I suggest publication after reviseing some minor defects.

 1. The sentence is obscure, what the “using existing methods” means? (Line 19-21)

 2. The subgraph titles in the manuscript is best to use editable text. (Fig.2 , 3, 11, and so on)

 3. I think it is better to unify the font size of words in the figures across the whole manuscript, for example, the font size of words in Fig.4 is rather too large compared with others.

 4.Please explain what the black solid circle and the red solid circle represent respectively ?(Figure 8)

 5. why use “CZ ” and “BZ” in the text, If they are an abbreviation, then the full name is? (Line 412-413)

Author Response

Thank you for your feedback. Below are responses to each of your comments:

1. The sentence is obscure, what the “using existing methods” means? (Line 19-21)

Response: Thank you very much for your suggestion. The “existing methods” refers to the classical method of extracting the channels, which can be performed with ArcGIS. And this sentence has been rewritten in the revised manuscript.

 “To solve this problem, this study proposes a method to identify and eliminate parallel channels extracted by classical methods.”

2. The subgraph titles in the manuscript is best to use editable text. (Fig.2 , 3, 11, and so on)

Response: Thank you for your suggestions. The manuscript has been revised in accordance with your recommendations.

3. I think it is better to unify the font size of words in the figures across the whole manuscript, for example, the font size of words in Fig.4 is rather too large compared with others.

Response: Thank you for your suggestion. Due to variations in the size of figures in the manuscript, achieving uniform font sizes across all figures is challenging. We will make efforts to adjust and address this issue as much as possible.

 4.Please explain what the black solid circle and the red solid circle represent respectively ?(Figure 8)

Response: The black solid circle represent the intersections of lines connecting second level channels and first level channels, while the red solid circle denote the DC-Point. We did not clearly indicate this in Figure 8 of the manuscript, and it has been revised accordingly.

5. why use “CZ ” and “BZ” in the text, If they are an abbreviation, then the full name is? (Line 412-413)

Response: Thank you for your suggestion. We defined these abbreviations using Chinese terminology, which indeed was not very logical. Therefore, we have corrected the nomenclature in the manuscript: "CZ" has been changed to "DV," with the full name "Difference Value," and "CZ" has been changed to "RV," with the full name "Ratio Value"

Reviewer 5 Report

Comments and Suggestions for Authors

The manuscript presents an algorithm to identify and eliminate  parallel channles from drainage networks derived from DEMs. Parallel channles widely exist in drainage networks if no special treatments are applied. The research is interesting and has great significance for quantitative geomorphic feature analysis. The content organization of the manuscript is reasonable, the experimental analysis is sufficient, and the language expression meets the publication requirements. So, I suggest that publish the manuscript after revising the following small questions.

1. The full name of the abbreviation only needs to appear once in the text. (L17, L44)

2.What the numbers in Figure 2 represent should be explained in the appropriate place of the Figure title or the text.

3. The revised channle netowrk in the first area seems to be discontinuous. Please check and confirm it. (Figure.19)

Author Response

Thank you for your feedback. Below are responses to each of your comments:

. The full name of the abbreviation only needs to appear once in the text. (L17, L44)

Response: Thank you for your suggestions. The manuscript has been revised in accordance with your recommendations.

2.What the numbers in Figure 2 represent should be explained in the appropriate place of the Figure title or the text.

Response: Thank you for your suggestions. The manuscript has been revised in accordance with your recommendations.

3. The revised channle netowrk in the first area seems to be discontinuous. Please check and confirm it. (Figure.19)

Response: Thank you for your suggestion.  We have re-examined the content, and the discontinuity in the channel network in the first region is a result of the compression caused by inserting images into Word.  In our subsequent submissions through the system, we will submit the images separately to address this issue.
